# Establishment of an In Vitro Model to Study Viral Infections of the Fish Intestinal Epithelium

**DOI:** 10.3390/cells12111531

**Published:** 2023-06-01

**Authors:** Guro Løkka, Amr A. A. Gamil, Øystein Evensen, Trond M. Kortner

**Affiliations:** Department of Paraclinical Sciences, Faculty of Veterinary Medicine, Norwegian University of Life Sciences, P.O. Box 5003, NO-1432 Ås, Norway; amr.gamil@nmbu.no (A.A.A.G.); oystein.evensen@nmbu.no (Ø.E.); trond.kortner@nmbu.no (T.M.K.)

**Keywords:** intestinal epithelial cells, gut barrier function, rainbow trout, in vitro models, virus, RTgutGC, IPNV, SAV3, ISAV

## Abstract

Viral infections are still a major concern for the aquaculture industry. For salmonid fish, even though breeding strategies and vaccine development have reduced disease outbreaks, viral diseases remain among the main challenges having a negative impact on the welfare of fish and causing massive economic losses for the industry. The main entry port for viruses into the fish is through mucosal surfaces including that of the gastrointestinal tract. The contradictory functions of this surface, both creating a barrier towards the external environment and at the same time being responsible for the uptake of nutrients and ion/water regulation make it particularly vulnerable. The connection between dietary components and viral infections in fish has been poorly investigated and until now, a fish intestinal in vitro model to investigate virus–host interactions has been lacking. Here, we established the permissiveness of the rainbow trout intestinal cell line RTgutGC towards the important salmonid viruses—infectious pancreatic necrosis virus (IPNV), salmonid alphavirus (subtype 3, SAV3) and infectious salmon anemia virus (ISAV)—and explored the infection mechanisms of the three different viruses in these cells at different virus to cell ratios. Cytopathic effect (CPE), virus replication in the RTgutGC cells, antiviral cell responses and viral effects on the barrier permeability of polarized cells were investigated. We found that all virus species infected and replicated in RTgutGC cells, although with different replication kinetics and ability to induce CPE and host responses. The onset and progression of CPE was more rapid at high multiplicity of infection (MOI) for IPNV and SAV3 while the opposite was true of ISAV. A positive correlation between the MOI used and the induction of antiviral responses was observed for IPNV while a negative correlation was detected for SAV3. Viral infections compromised barrier integrity at early time points prior to observations of CPE microscopically. Further, the replication of IPNV and ISAV had a more pronounced effect on barrier function than SAV3. The in vitro infection model established herein can thus provide a novel tool to generate knowledge about the infection pathways and mechanisms used to surpass the intestinal epithelium in salmonid fish, and to study how a virus can potentially compromise gut epithelial barrier functions.

## 1. Introduction

Viral infections still cause large economic losses for the aquaculture industry and impose negative impacts on the health and welfare of fish. For Atlantic salmon and rainbow trout, the selective breeding of fish strains with a high resistance towards some important virus diseases such as infectious pancreas necrosis (IPN) and more recently pancreas disease (PD) has been employed to reduce the disease incidences [1,2]. Further, several viral vaccines are available, either as part of multivalent vaccines or as individual targeted vaccines such as the DNA vaccine against PD. However, despite the wide use of the viral vaccines, the number of clinical outbreaks after vaccination was implemented did not show a marked decline for IPN and PD, and viral infections were still among the top 10 challenges in the Norwegian fish production industry in 2022 [3].

The mucosal surfaces of fish, including those of the skin, gills and gastrointestinal tract, create barriers towards the external environment and are primary routes for the entry of infectious agents and other antigens into the host. At the same time, the intestinal surface is responsible for nutrient absorption and fluid homeostasis. To meet the requirements for these two contradictory functions, the intestine has developed a selective and complicated semi-permeable barrier. The movement of molecules (including fluid exchange, intestinal nutrient uptake and antigen sampling) over the intestinal epithelium can be mediated by paracellular or transcellular pathways [4]. Paracellular transport is rate-limited by the tight junctions of the adjacent epithelial cells and both the size and charge of the molecules are decisive for the transport. The transcellular pathway across the cell membranes may be both receptor-mediated or through non-specific hydrophobic interactions between the cell membrane and the antigen [5]. Dietary, pathological and microbial factors may affect both paracellular and transcellular transport capacities [6]. In bony fish, also called teleost fish, previous studies have shown that antigen uptake and transport across the intestinal epithelial barrier mainly occur in the distal intestine (reviewed in [7]).

Knowledge about the relationships between nutrition, immune responses and viral diseases in aquaculture is very limited, not least due to a lack of targeted research tools. Various feed additives included in functional feeds in the fish industry are branded not only by their nutritional value, but also based on their claimed health-promoting and disease-preventing properties. To evaluate the effects of diet on disease resistance, challenge studies with live fish exposed to pathogens and mortality/morbidity as the end point are considered the gold standard. However, due to increased animal welfare awareness, reduction of the use of animals have been promoted and the generation of new in vitro models to replace live animal trials has consequently become one of the main prerequisites in animal research [8,9]. As an in vitro alternative, intestinal-derived cell lines may serve as models to initially test and screen for the effect of different substances on microbial infections at the cellular level before conducting animal experiments. In a previous study, we used the RTgutGC cell line [10] as a model for studying gut immune function and the effects of functional feed ingredients [11]. The RTgutGC cell line is an immortal cell line originating from the distal intestine of an adult female rainbow trout and has been described as having an epithelial-like morphology [10]. Since their initial isolation, RTgutGC cells have been well characterized and routinely grown as monolayers both on conventional plates and on porous membranes in transwell systems, where the cells create a polarized barrier between the apical (corresponding to the intestinal lumen) and the basolateral (corresponding to the blood flow) compartments [11,12]. Previous studies on RTgutGC cells have demonstrated tight junction and desmosome formation between adjacent cells, the development of transepithelial electrical resistance and polarization over time as well as the expression of epithelial and brush border characteristics [11,12,13,14,15]. Consequently, the RTgutGC cell line has as such been proposed as a physiologically adequate fish intestinal epithelial model equivalent to the human intestinal epithelial cell line Caco-2 and has been used for studying intestinal immune function, eco-toxicology and nutrition [11,16,17,18].

Although RTgutGC cells have been used to study functional ingredients, no previous studies were conducted to study virus infection using these cells and their permissibility to virus infections remains unknown. In the present work, the aim was to establish an in vitro infection model for studying the interaction between salmonid viral pathogens and the intestinal cells using the RTgutGC cell line. The selected viruses were infectious pancreatic necrosis virus (IPNV), salmonid alphavirus 3 (SAV3) causing PD and infectious salmon anemia virus (ISAV). While IPNV infections have been under control in Norway in recent years, PD is frequently diagnosed in Norwegian aquaculture and is listed as a fish infectious disease by the World Organization of Animal Health (WOAH) [3]. Outbreaks of ISAV is moreover a persistent problem in Norwegian farms and will result in strict measures being imposed on affected farms, as the disease is listed both in the EU and by the WOAH [3].

The in vitro infection model established herein can provide a novel tool to generate knowledge about the infection pathways and mechanisms used to surpass the intestinal epithelium in salmonid fish. This system may further be used for predicting the effects of functional feed ingredients on pathogen infections, disease resistance as well as gut barrier functions and health in salmonid fish.

## 2. Materials and Methods

### 2.1. Cell Culture

Rainbow trout intestinal epithelial cells RTgutGC [10] (kindly provided by Prof. Kristin Schirmer, Eawag, Dübendorf, Switzerland) were cultivated in 75 cm^2^ tissue culture flasks (TPP, Trasadingen, Switzerland) with L-15/FBS, i.e., Leibovitz’s L-15 medium without Phenol Red (Gibco, Invitrogen, Waltham, MA, USA) supplemented with 5% fetal bovine serum (FBS) (Eurobio Scientific, Les Ulis, France). Cells were incubated at 20 °C under a normal atmosphere. Every 7 to 10 days, at approximately 80–90% confluence, the cells were split in a 1:2 ratio using trypsin (0.25% in phosphate buffered saline (PBS) w/o Ca^2+^ and Mg^2+^; Biowest, Nuallé, France). The cells were routinely tested to be free of mycoplasma. Cell line passages between 87 and 116 were used in the current experiments, well within the range of the 150 sub-passages that have been tested and described to maintain unchanged characteristics [10].

### 2.2. Virus Propagation

To obtain adequate amounts of the viruses for the different assays performed, a highly virulent IPNV isolate (TA strain, NVI-015) [19], SAV3 (isolate H10) [20] and a field isolate, F8, of ISAV, with the same deletion pattern in the hemagglutinin-esterase gene observed in the Norwegian HPR 5 isolate T152/09 (Genbank JN711086) [21] were propagated in rainbow trout gonad-2 (RTG2) cells, Chinook salmon heart-1 (CHH1) cells and Atlantic salmon kidney-2 (ASK2) cells, respectively. Cells of the respective cell lines were grown at 20 °C in L-15 medium (Gibco) supplemented with 10% FBS until 80% confluency before they were inoculated with the different viruses. Before virus inoculation, the media was replaced with L-15 medium supplemented with 1% FBS and 50 µg/mL gentamycin (Invitrogen). Inoculated cells were incubated at 15 °C until the full cytopathic effect (CPE) was reached. Supernatants containing the different viruses were then harvested, centrifuged at 2500 rpm for 10 min and filtered (0.22 µm syringe filters; Whatman™, VWR, PA, USA) before further use. The titer of the viruses was estimated by limiting dilution in 96-well plates (Falcon, Bedford, MA, USA) containing 80–90% confluent Chinook salmon embryonic (CHSE), CHH1 and ASK2 cells for IPNV, SAV3 and ISAV, respectively. The 50% tissue culture infective doses (TCID_50_) were calculated using Kärber’s method [22].

### 2.3. Virus Infection Assays and Assessment of CPE in RTgutGC Cells

To assess the permissiveness of the RTgutGC cell line to the selected viruses, RTgutGC cells were seeded in 24-well plates (Greiner Bio-One, Kremsmünster, Austria) at a density of 5 × 10^4^ cells in 1 mL of L-15/FBS (~26,300 cells/cm^2^) and grown until 80% confluence at 20 °C before the virus infection. The cells were inoculated with the virus at a multiplicity of infection (MOI) of 0.1, 1 or 10 virus particles/cells in L-15 medium containing 1% FBS and 50 µg/mL gentamycin and held at 15 °C. Inoculations were performed in triplicate wells and the experiment was repeated twice with different passages of RTgutGC cells. Inoculated cells were observed, and phase contrast images captured using an inverted Olympus IX81 microscope with a ColorView Soft Imaging System (Olympus Life Science Solutions, Tokyo, Japan) at different times post-infection until full CPE was reached.

### 2.4. Immunofluorescence Antibody Technique (IFAT)

For the detection of virus inside cells, IFAT was performed at 2 days post-infection (dpi) and 6/7 dpi for the three MOIs. RTgutGC cells seeded in 24-well plates and inoculated with the virus as described above were fixed for 20 min with 4% paraformaldehyde (Sigma-Aldrich, Merck KGaA, Darmstadt, Germany) in PBS (pH 7.4). Cells infected with SAV3 were permeabilized with ice cold acetone/methanol (1:1) for 10 min at 20 °C due to difficulties with getting this antibody to permeate the cells. For IPNV and ISAV, the cells were permeabilized on ice with 0.1% Triton X-100 (Sigma-Aldrich) in PBS for 5 min. For all of the three viruses, cells were blocked with 5% bovine serum albumin (BSA) (Sigma-Aldrich) in PBS for 20 min. Primary antibodies diluted in 2.5% BSA in PBS were then added after discarding the blocking medium and allowed to incubate overnight at 4 °C. The primary antibodies applied were a polyclonal antibody towards IPNV serotype Sp. (K95 (1:1000) [23]), SAV3-antibody (anti-E2 (1:100) [24]) and monoclonal antibody against ISAV ((1:1000) Aquatic diagnostics, Scotland, UK). The following day, cells were incubated with secondary antibodies (Alexa Fluor™ 488 goat anti-rabbit IgG (H + L) (Invitrogen) or goat anti-mouse for IPNV/SAV3 and ISAV, respectively) at dilution 1:1000 in 2.5% BSA for 30 min at room temperature, before being counterstained with nuclei-specific Hoechst fluorescent dye (5 µg/mL; Thermo Fisher Scientific, Waltham, MA, USA). Between all steps, except after blocking, cells were washed twice with PBS for 5 min. Cells were kept in PBS before visualization with an Olympus IX81 microscope with a ColorView Soft Imaging System (Olympus Life Science Solutions).

### 2.5. Quantitative Real-Time PCR (qPCR) Analyses of Virus Replication and Cellular Responses

qPCR was used to quantify the level of viral mRNA as well as the expression of antiviral response genes, namely interferon alpha (*ifn-a*), myxovirus resistance protein 1 (*mx-1*), protein kinase R (*pkr*), interferon regulatory factor 9 (*irf-9*) and the pro-inflammatory gene tumor necrosis factor alpha (*tnf-α*). RTgutGC cells were seeded in 12-well plates (Greiner Bio-One) at a density of 5 × 10^4^ cells/mL in 2 mL of L-15/FBS (~25,600 cells/cm^2^). Cells were grown at 20 °C for 48 h until 80% confluence was reached before being inoculated with IPNV, SAV3 or ISAV. The virus was added to the cells at MOI 0.1, 1 or 10 diluted in L-15 medium containing 1% FBS and 50 µg/mL gentamycin. Separate plates were used for each virus to avoid cross-contamination and infections were performed in triplicate wells while 2–3 uninfected control wells were included in each plate. After inoculation, the plates were incubated at 15 °C until harvesting. The time points for cell harvesting were determined based on results from the CPE assays. For IPNV, cells were harvested at 3 dpi while for SAV3 and ISAV, cells were collected at 5 dpi. Samples were thus taken at an early stage of the infection for all three viruses. At the respective time points, the supernatants were removed from the cells and cells were lysed with TRIzol^®^ (Thermo Fisher Scientific) (1 mL per well). Cell lysates were collected in 2 mL microcentrifuge tubes with screw caps (Thermo Fischer Scientific) and saved at −20 °C until further processed. Total RNA was extracted using the PureLink^®^ RNA Mini Kit (Thermo Fisher Scientific) as described by the manufacturer, including a DNase treatment step. After extraction, the RNA integrity was verified by the Agilent 2200 TapeStation system in combination with an RNA ScreenTape (Agilent Technologies, Santa Clara, CA, USA), and the samples had an average RNA integrity number (RIN) of 7.6. On the other hand, RNA purity and concentrations were measured using the Epoch Microplate Spectrophotometer (Agilent Technologies). Total RNA was subsequently stored at −80 °C until cDNA was synthesized using 200 ng RNA and the Superscript IV VILO mastermix (Invitrogen, Waltham, MA, USA) in 20 µL reactions according to the manufacturer’s protocol. Negative controls were also prepared in parallel by omitting the RNA or enzyme. The obtained cDNA was diluted 1:5 and stored at −20 °C. The real time PCR analyses were performed using the LightCycler 96 (Roche Diagnostics, Rotkreuz, Switzerland). A 10 μL reaction mix containing 2 μL PCR-grade water, 2 μL of 1:5 diluted cDNA template (corresponding to 4 ng total RNA), 5 μL of LightCycler 480 SYBR Green I Master (Roche Diagnostics) and 0.5 μL (final concentration 500 nM) of each forward and reverse primer was used for each sample and the samples were assayed in duplicates. A no template control (NTC) was also included. The sequences of primers used in the reactions are provided in Appendix A. The three-step qPCR program included an enzyme activation step at 95 °C (5 min) and 45 cycles of 95 °C (10 s), 60 °C (10 s) and 72 °C (15 s). The results were analyzed by the ΔΔcq relative quantification method [25] with *beta-actin* as the reference gene. 

### 2.6. Permeability Test with Fluorescent Bovine Serum Albumin (BSA)

To assess the effect of viral infections on the rainbow trout intestinal epithelial barrier function, an in vitro barrier permeability test with fluorescent BSA was performed. RTgutGC cells were seeded in the apical compartment of well inserts with permeable membranes (pore size 3 µm; pore density 0.6 × 10^6^/cm^2^) in 6-well plates (Greiner Bio-One), at a concentration of 8 × 10^4^ cells/mL in 3.5 mL of L-15/FBS (~61,900 cells/cm^2^). In the basolateral compartment of the insert well system, 3.5 mL of L-15/FBS without cells was added. After 1 week cultivation at 20 °C, medium was replaced in both apical and basolateral chambers with fresh L-15/FBS and thereafter once a week until 28 days (4 weeks) after seeding. Cells were then inoculated with a virus (IPNV and SAV3: MOI 1; ISAV: MOI 0.1) in L-15 medium supplied with 1% FBS and 50 µg/mL gentamycin. Four replicate wells were treated for each virus, and control wells without a virus (c1) as well as blank wells without cells were included in the set-up. The permeation of fluorescent BSA (size 66 kDa) across the epithelial cells was used to assess the barrier function. After 6 h incubation with a virus at 15 °C, 25 µg of BSA conjugated to Alexa Fluor™ 488 (Thermo Fisher Scientific) was added to the apical side of each insert. Control wells without a virus where BSA-Alexa Fluor™ 488 was not added (c2) were also run in parallel and were used to measure background fluorescence. Plates were incubated at 15 °C and samples were taken from the basolateral side at different time points starting at 1 h and up until 7 days (1, 2, 16, 64, 88 and 160 h after addition of BSA or 7, 8, 22, 70, 94 and 166 h after infection). Samples (100 µL) were pipetted directly into a Nunc™ Maxisorp™ flat-bottom 96-well plate (Thermo Fisher Scientific) and the fluorescence signal (excitation 490; emission 525) was read using a Cytation3 plate reader (Agilent Technologies). This experiment was repeated twice.

### 2.7. Statistical Analyses

GraphPad Prism 7.03. (GraphPad Software, San Diego, CA, USA) was used for the statistical analyses and graphical illustrations of the data from the gene expression and permeability assays. A normal distribution of the data was assessed using a Shapiro–Wilk normality test and a Brown–Forsythe test was applied to test for significant differences between standard deviations. Data that did not pass these tests were log-transformed. For gene expression data, significant differences between the treatments were investigated using a one-way ANOVA test and comparisons between the different MOI treatments were performed with Tukey’s multiple comparison tests. Linear regression analyses were also performed for gene expression results when a straight line was the best fitted curve for the data. Correlations between the viral replication and expression levels of antiviral genes were assessed with Pearson’s correlation where r > 0.70 was considered a strong positive correlation. For the permeability assay, a two-way ANOVA analysis with Tukey’s multiple comparison test was performed with time and treatment set as the two factors. In addition, comparisons between uninfected wells and virus-treated wells for each time point were run using Dunnett’s multiple comparisons test. All data are presented as mean ± SEM.

## 3. Results

### 3.1. Assessment of CPE

The ability of the three selected viruses to replicate and induce CPE in RTgutGC cells was assessed using conventional 24-well plates. All three viruses induced CPE while no CPE was observed in uninfected control cells (Figure 1).

Inoculation with IPNV induced a distinct CPE in the RTgutGC cells by 1 dpi, exemplified by the appearance of virus vacuoles inside cells, cells detaching from each other and the extracellular accumulation of cell debris (Figure 1D). As the IPNV infection progressed, cells were lysed by the virus and the cellular density in the well thus markedly decreased, leaving only cell debris and occasional scattered elongated cells (Figure 1E,F). On the other hand, SAV3 infection did not induce as prominent CPE as IPNV, but slight cellular detachment, cell deformation and a modest increase in cell death were observed by 4 dpi (Figure 1G). As the SAV3 infection progressed, cellular detachment increased and rounded, apoptotic-like cells were prominent. In addition, dead cells were floating in the medium (Figure 1H,I). ISAV infection was characterized by increased rounded apoptotic-like dead cells with infection progression, first observed by 4 dpi, and the CPE was similar to the one observed in SAV3 infected cells (Figure 1J–L).

A difference was observed between the occurrence of CPE relative to the MOIs used. For IPNV, increasing the MOI resulted in an earlier onset of CPE (Figure 2). At MOI 10, CPE was observed at 1 dpi, while for MOI 1 and 0.1, CPE was first observed at 2 dpi and at 3 dpi, respectively. IPNV infection progressed rapidly reaching full CPE at 6 dpi for MOI 10, and by 7 dpi at MOIs 1 and 0.1. At 3 dpi, the samples for qPCR were collected and CPE is shown in Appendix A. For SAV3, CPE was first observed at 4 dpi for MOIs 10 and 1, and by 5 dpi for MOI 0.1 (Figure 2). The infection was more protracted for SAV3 than for IPNV, with full CPE reached at 13 dpi for MOI 10 and 1, and by 14 dpi for MOI 0.1. For SAV3, at 5 dpi when the samples for qPCR were collected, CPE is shown in Appendix A. In contrast to IPNV and SAV3, ISAV induced CPE earlier for MOI 0.1 and 1 (at 4 dpi) than for MOI 10 (at 7 dpi) (Figure 2). Correspondingly, full CPE was reached first for MOI 0.1 at 10 dpi, by 12 dpi for MOI 1 and 13 dpi for MOI 10. qPCR samples were collected at 5 dpi with CPE documented in Figure 3.

### 3.2. Detection of Virus Proteins in Infected RTgutGC Using IFAT

Immunofluorescence antibody staining was used to demonstrate a viral infection in RTgutGC cells at different dpi and at different MOIs. While no viral antigens were detected in the control cells for any of the viral antibodies, positive staining was detected in the infected cells. IFAT staining for IPNV, at 2 dpi and at MOI 1 and 10, showed strong staining in more than 50% of the cells (Figure 4A,C). At MOI 10, cell density was reduced (as a consequence of infection), and cell morphology was affected by the virus infection with more elongated cells being observed. At 7 dpi, for MOIs 1 and 10, cells were mostly lysed, and the remaining cells stained positive (Figure 4B,D).

For SAV3, the IFAT signal was weaker compared to the two other viruses and most cells showed a faint signal with only a few cells with distinct staining (Figure 5). Most cells stained positive at 6 dpi at MOI 10 and cell density was decreased by 14 dpi, as a consequence of CPE.

For ISAV, at 2 dpi and MOI 0.1 and 1, most cells showed strong, cytoplasmic staining for virus antigens, while in wells infected at MOI 10, hardly any virus-infected cells were observed (Figure 6A,C,E). In contrast, at 7 dpi, most ISAV-positive cells were observed in wells infected at MOI 10, while at MOI of 0.1 and 1, virus-positive cells were scarcer, and cells showed severe CPE and were detached (Figure 6B,D,F). In addition, the staining pattern was localized at 7 dpi in cells infected at MOI 0.1 and 1.

### 3.3. Correlation between Viral Replication and Induction of Antiviral Responses

To understand the molecular basis of the differences observed in CPE and virus replication, qPCR was employed. The viral replication rates within cells were quantified and compared to the expression levels of antiviral and pro-inflammatory response genes. Viral replication was detected in RTgutGC cells infected with all three viruses, while not in uninfected control cells. Viral loads in infected RTgutGC cells varied with the MOI used for inoculation, but responses differed for the three viruses (Figure 7). For IPNV and samples taken at 3 dpi, the viral mRNA (i.e., reflecting the virus replication within the cells) increased with an increasing MOI (MOI 1 significantly higher than MOI 0.1 (*p* = 0.02) and MOI 10 significantly higher that both MOI 0.1 and MOI 1 (*p* < 0.0001)) (Figure 7A). Cells infected with MOI 0.1 had however a very low viral load and did not rise above uninfected control cells. The data were fitted to a linear regression curve, and the slope was significantly different from zero (*p* < 0.0001, not shown).

The induction of the antiviral and pro-inflammatory immune responses in IPNV infected cells showed a similar trend as described for virus replication (Figure 8) and there was a high correlation between the induction of the antiviral genes and the increase in viral mRNA levels (r > 0.9). For all investigated genes, cells infected with IPNV at MOI 10 had significantly higher expression levels (*p* < 0.0001) than cells infected with a lower MOI and uninfected control cells (Figure 8A). Antiviral and pro-inflammatory markers were not differently expressed in cells infected with IPNV at MOI 0.1 and 1 compared to control cells. The expression of all genes showed a linear increase with IPNV MOI (linear regression, *p* < 0.0001 for all genes, not shown).

For SAV3 (samples taken at 5 dpi), the virus yield was significantly highest in the cells infected with the highest MOI (*p* = 0.01 and *p* = 0.0005 compared to MOI 0.1 and MOI 1, respectively), while no statistically significant difference was observed between MOI 0.1 and 1 (Figure 7B). These results could also be fitted to a linear regression curve with a slope significantly different from zero (*p* = 0.0045, not shown). Unlike IPNV, the expression levels of the antiviral and pro-inflammatory genes in SAV3-infected cells did not reflect the increase in MOI or the viral mRNA levels (0.3 < r < 0.6). Rather, the general trend was an inverse correlation between the responses induced and the MOI used as well as the virus mRNA levels. Accordingly, cells infected at MOI 0.1 had a significantly higher expression level for all genes investigated compared to 1 and 10 MOI (Figure 8B), and the expression of the investigated genes was strongly positively correlated to each other (r > 0.7). No statistically significant differences were detected between the expression of the different genes in 1 and 10 MOI-infected cells although the levels were generally lower in the latter.

In contrast, for ISAV (samples collected at 5 dpi), the viral mRNA levels were inversely related to the MOI used for infection, meaning that the cells infected at MOI 0.1 showed the highest viral yield (Figure 7C) while cells infected with ISAV at MOI 10 was the lowest. Cells infected at 0.1 and 1 MOI were highly significant in viral mRNA levels compared to cells infected at MOI 10 (*p* < 0.0001 for both, but MOI 0.1 and MOI 1 were not significant from each other), while the viral mRNA levels from the cells infected with ISAV at MOI 10 did not differ significantly from uninfected cells (*p* = 0.9). These data did not fit a straight line but had a more curved regression. The correlation between viral replication and gene expression was strongly positive only for ifn-a (r = 0.94) and tnf-α (r = 0.7), as these genes showed the highest expression at 0.1 MOI (Figure 8C), and they were strongly positively correlated to each other (r = 0.8). Interestingly, for interferon-induced genes mx-1, pkr and irf-9, the expression levels were equally high in cells infected at MOI 10 as in cells infected at MOI 0.1. The interferon-induced genes were moreover strongly positively correlated to each other (r > 0.8).

### 3.4. Effects of Virus Infection on Epithelial Barrier Function

The permeability of the intestinal epithelial barrier after viral infections was assessed by growing RTgutGC cells on a porous membrane in a two-compartment transwell system and measuring the diffusion of fluorescent BSA across the barrier (Figure 9). Samples taken from the basolateral side of wells at 7 and 8 h post-virus infection (hpi: i.e., 1 and 2 h after addition of BSA, respectively) showed low fluorescence signal and there was no difference between the control wells and wells infected with a virus. After 22 hpi, however, samples taken from wells infected with IPNV and ISAV showed significantly higher fluorescence signal than uninfected wells, indicating that the virus had weakened the intestinal epithelial barrier. The same result was obtained from samples taken 3 days (70 h) after infection. After 7 days (166 h), infection with all three viruses gave significantly higher fluorescence signal compared to uninfected wells. The two-way ANOVA analysis showed that the permeability of the barrier was significantly affected both by time after infection and by treatment (*p* < 0.0001). Comparing between the different treatments, only IPNV and ISAV differed significantly from the control.

## 4. Discussion

In this study, in vitro fish intestinal infection models were established using the RTgutGC cell line and salmonid viruses IPNV, SAV3 and ISAV. These viruses had different infection dynamics and induced varying responses in the RTgutGC cells depending on the infection dose. IPNV and ISAV infections had more severe effects on the polarized in vitro intestinal barrier than SAV3 and compromised the barrier function prior to visible cytopathogenic effects in the cells. This indicates that the junctional complexes between the cells were weakened shortly after infection, and this probably constitute an important entry passage for these viruses into the host.

With the current in vitro intestinal infection models in place, the virus interaction with the salmonid intestinal barrier may be explored further without the unnecessary use of live animals. Due to animal welfare concerns, the use of animals in experiments has come under scrutiny in recent years and has resulted in more stringent criteria and regulations. The three Rs (3R) concept which promotes the reduction, refinement, and replacement of animals in experiments has consequently been brought to the fore. In some cases, however, such as when studying the virus–host interaction, the use of animals is difficult to avoid; but in vitro models can be used to establish basic knowledge before the animal experiments are performed. The intestinal barrier is one of the main routes of entry for many viruses and the interplay between the intestinal epithelium and viruses is therefore an important field of study. In vitro models to study the virus interaction with intestinal cells are in place in higher vertebrates including intestinal organoid models derived from self-renewing and self-organizing stem cells [8,26], but the same is not true for fish. In salmonid fish, the intestinal stem cell niche has just started to be unraveled indicating a different organization and different stem cell markers compared to mammals [27], and the isolation and culturing of fish intestinal stem cells have so far not been successful. However, the RTgutGC has recently emerged as a cell line with the potential to be used as an intestinal cell line in vitro models for salmonids and has been used mainly in toxicity studies [17,28,29,30,31,32,33]. Recently, we have used this cell line to study the interaction between functional ingredients and the intestine and have established an in vitro assay to mimic and measure the effect on intestinal permeability [11]. When RTgutGC cells are grown on a permeable membrane insert that separates an apical (mimicking the intestinal lumen) and a basolateral (mimicking the venous blood) compartment, both compartments are easily accessible for continuous sampling to evaluate the permeability across the membrane and thus the barrier function. Although the recent mammalian organoid models better mimic the 3D structure of the intestine, the cystic and spherical structure of these organoids have limitations in accessing the luminal compartment that will be buried in a Matrigel mass. Here, we have thus expanded the use of the transwell model and used the RTgutGC cell line and the permeability assay to establish an in vitro system to assess the interplay between intestinal cells and viruses. Our data show that the RTgutGC cells are permissible to IPNV, SAV3 and ISAV and that the in vitro permeability assay using a transwell system can be applied to evaluate the impact of these viruses on the intestinal cell function.

To establish the in vitro fish intestinal epithelial viral infection model, we first tested the permissibility and replication dynamics of three of the important salmonid viruses, namely IPNV, SAV3 and ISAV, in the RTgutGC cell line. The replication of these viruses in RTgutGC cells has not been investigated previously although viral hemorrhagic septicemia virus, chum salmon reovirus and frog virus 3 were shown to replicate in these cells [34]. We found that all the three viruses infect and replicate in the RTgutGC cells but with different replication kinetics and the ability to induce CPE and host responses. Among the three viruses, IPNV infection induced the most prominent CPE in the RTgutGC cells, exhibited the earliest onset of CPE (2 dpi) and reached full CPE faster. Corresponding results were reported in salmon TO cells (head kidney), where infection with IPNV induced CPE already at 1 dpi (MOI not defined) and earlier than for ISAV and SAV (strain not defined) [35]. The high ability of IPNV to infect and replicate in the RTgutGC cell line is not surprising as IPNV has been found to propagate in several cell lines of epithelial origin and to replicate in vivo in the intestinal cells of salmonid fish [36]. Further, the translocation of IPNV across the rainbow trout intestinal mucosa has been demonstrated in vivo and ex vivo by the Ussing chamber technique, providing evidence for the gastrointestinal tract as an entry route for IPNV [37,38]. The high replication of IPNV coexisted with the high expression of the antiviral genes ifn-a, mx-1 and pkr, especially when a high MOI was used for infection. A similar strong ifn-a response has been described earlier in CHSE cells when infected with different IPNV strains (MOI 20) including the TA strain used herein [39]. Our data therefore support the view that IPNV is able to evade the antiviral immune responses and/or use it to its advantage [40,41,42].

On the other hand, the SAV3 infection gave the highest virus titer among the three viruses studied (10^6^). Still, infection with SAV3 in the RTgutGC cells did not induce as pronounced CPE as IPNV, at least not at the early time of infection. This is in line with previous observations in salmon TO cells where the CPE from SAV (strain/type not defined) was not easily observable except for small plaque late in the infection period even though virus transcription and immunostaining proved permissiveness [35]. In contrast, in other studies with TO cells, infection with SAV2 and SAV3 isolates induced prominent CPE [24,43], indicating that the different SAV subtypes have different virulence. In addition, IFAT staining was weaker for SAV3 than for the other two viruses, but this may have been due to a suboptimal antibody protocol for virus detection. For SAV, probable infection routes are through the gills and the intestine [44]. However, in vitro, SAV2 could only replicate at low levels in the rainbow trout gill epithelial cell line, RTgill-W1 [45]. The finding that SAV3 replicates at high levels in the RTgutGC cells may point to the intestinal epithelial cells being a more important entry route into the host for this virus strain than earlier assumed but this requires further investigation. Similar to IPNV, the RTgutGC cells elicited strong antiviral and pro-inflammatory responses to the SAV3 infection. Interestingly, all investigated genes had the highest expression levels when infected at MOI 0.1 while viral replication of the cells was highest at MOI 10. High levels of intracellular SAV3 virus may thus possibly block or suppress expression of antiviral responses in the RTgutGC cells. Both mammalian and fish alphaviruses including SAV have been described to suppress or evade host interferon responses to escape the innate immune system of the host [46] and this could be related to protein shutdown [24]. A comparative study of SAV2 infection in TO and SHK-1 cells showed higher expression of the interferon-induced gene mx in SHK-1 cells concomitant with a lesser ability to induce CPE compared to TO cells, indicating that the ability of the virus to interfere with the cell’s interferon signaling pathway is important for intracellular SAV replication and the ability to induce cytopathogenicity [43].

For ISAV, an inverse relationship between the MOI used for infection and cell responses was observed in terms of CPE onset as well as viral replication levels by qPCR and IFAT staining. Due to a low titer of the propagated ISAV (10^5^), a large volume of viral solution was necessary to obtain MOI 10. It is therefore possible that a high interferon level in the ISAV inoculate added to the cells may have had an impact on the results. Another possibility is that high virus levels result in the rapid detection and immune sensing of the virus infection leading to rapid development of high-level antiviral responses. However, the expression levels of mx-1, pkr and irf-9, were equally high in cells infected at MOI 0.1 and 10, which make the latter explanation unlikely. The primary entry port for ISAV into the host was first assumed to be the gills [47], but entry through multiple mucosal surfaces including the pectoral fin, skin and gastrointestinal tract has recently been demonstrated [48]. After entry into the host, the virus establishes in endothelial cells lining the blood vessels as the primary target of the infection [48]. The 4-O-acetylated sialic acid receptor has been demonstrated as essential for an ISAV infection of the cells by binding the virus, and this receptor has been found in the epithelial cells of both gills and the distal intestine [49,50], although histochemical labelling failed to reveal ISAV uptake by gastrointestinal epithelial cells [51]. The findings of this study together with the presence of the sialic acid receptor in the intestinal epithelium would support that ISAV can enter and replicate in the salmonid intestinal epithelial cells but this warrants further investigation. The RTgutGC cells can be a valuable tool with this regard.

All three viruses induced typical antiviral responses, i.e., the induction of interferon and pro-inflammatory cytokines in the RTgutGC cells. The innate immune response against viruses is typically triggered by pathogen-associated molecular patterns (PAMPs) such as dsRNA binding to host pattern–recognition receptors (PRRs) which triggers intracellular signaling pathways that result in the induction of interferons and pro-inflammatory cytokines [52]. The interferon response in combination with interferon stimulated genes, such as mx and pkr, work together to limit viral replication [53]. In the RTgutGC cell line, it has been shown earlier that the cells responded to extracellular viral dsRNA and Poly(I:C) stimuli by up-regulation of the interferon and interferon-stimulated genes, although with a different strength and timing between the different viral stimuli [15,54,55]. However, the presence and molecular characterization of the different intracellular and extracellular sensors have not been investigated in detail and may be required in future studies addressing the detailed virus–host interaction.

In the current study, a barrier was created when growing the RTgutGC cells for four weeks on the permeable membrane in a transwell system demonstrated by uninfected control wells having a stable and low fluorescent signal over time. After viral infections, the barrier was significantly affected, as shown by the increased translocation of fluorescent BSA across the barrier compared to uninfected control wells. This was first detected at 22 h post-infection for IPNV (MOI 1) and ISAV (MOI 0.1) and after 7 days for SAV3 (MOI 1). For ISAV MOI 0.1, visible CPE was not observed until 4 dpi (96 h) while for IPNV MOI 1, it was observed after 24 h. The permeability results thus indicate that although IPNV and ISAV was not affecting the morphology of the cells at 22 h after infection, the barrier function of the epithelium was still compromised by the virus, probably by changes in the tight junctional complexes between the cells. By affecting the tight junctional complexes, and making the epithelium more leaky, viral molecules can more easily access the host. It has previously been shown that stimulation with Poly(I:C) mimics viral antigens, affecting the expression of tight junction gene zo-1 in RTgutGC cells [15] suggesting that immune responses may be involved in affecting the tight junctions. Changes induced in permeability at later time points can be attributed to the cytopathological effects of the virus load, with the subsequent loss of cell monolayer.

The RTgutGC cells originate from the distal gut portion of the rainbow trout, which may give reason for the susceptibility of these cells to all three viruses. In salmonid fish, the distal intestine is regarded as the intestinal segment with function in antigen uptake (reviewed in [7]). Fish enterocytes themselves have been proposed as responsible for antigen sampling in contrast to the specialized M cells that recognize and take up antigens from the intestinal lumen in mammals [56]. It is therefore possible that more anterior/proximal regions of the gastrointestinal tract do not share the same susceptibility towards infection for these viruses. A support for this theory is the observation that the 4-O-acetylated sialic acid receptor, that binds ISAV, was only detected in luminal epithelial cells in the distal portion of the salmon intestine, and not in the pyloric region [50]. Until recently, immortal cell lines from the proximal or mid-region of the salmonid intestine have not been available for in vitro studies. Pasquariello and coworkers recently described, however, an established cell line from the proximal intestine of rainbow trout [14]. Further studies should therefore be performed with this cell line to study the possible differences between proximal and distal intestine concerning virus permissiveness.

## 5. Conclusions

The results presented in this study can be considered as the first step towards developing an in vitro model to study virus interactions with the salmonid intestinal epithelium using the well-established RTgutGC cell line. Here, we established the permissiveness of the RTgutGC cells to three important viruses of the salmonid industry, namely IPNV, SAV3 and ISAV, and demonstrated the infection mechanisms of these viruses at different MOIs. Permeability assays showed that the epithelial barrier function was negatively affected by the viral infections before CPE could be detected, and that IPNV and ISAV had a more severe effect on the barrier function compared to SAV3. The reported effect on intestinal permeability is interesting and more studies to understand the underlying mechanisms are required. A better characterization of the RTgutGC cell line in terms of the presence and molecular structure of the different immune sensors in addition to the responses induced by different stimuli will help to further develop this model.

## Figures and Tables

**Figure 1 cells-12-01531-f001:**
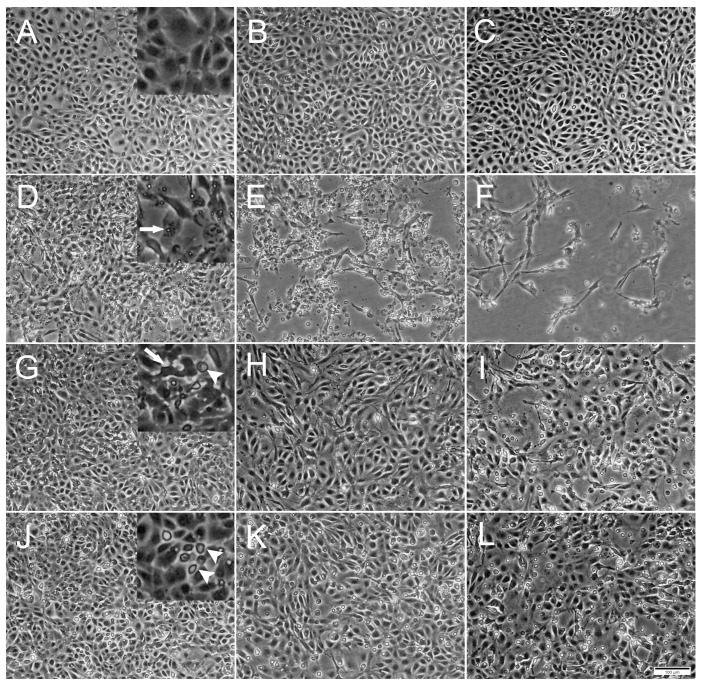
Phase contrast microscopical images of the development of cytopathic effects (CPE) in RTgutGC cells over time. (**A**–**C**) Uninfected cells incubated with L-15/1% FBS medium only, at 2-, 4- and 7-days post infection (dpi), respectively. Detail in (**A**) shows confluence of the cell layer. (**D**–**F**) Cells infected with IPNV at 2, 4 and 7 dpi. Detail in (**D**) shows virus vacuoles inside cell (arrow). (**G**–**I**) Cells infected with SAV3 at 4, 7 and 12 dpi. Detail in (**G**) shows deformed cell (arrow) detached from surrounding cells and rounded dead cell (arrowhead). (**J**–**L**) Cells infected with ISAV at 4, 7 and 12 dpi. Detail in (**J**) shows dead rounded cells (arrowheads). Multiplicity of infection (MOI) = 1 for all three viruses. Scale bar 100 µm.

**Figure 2 cells-12-01531-f002:**
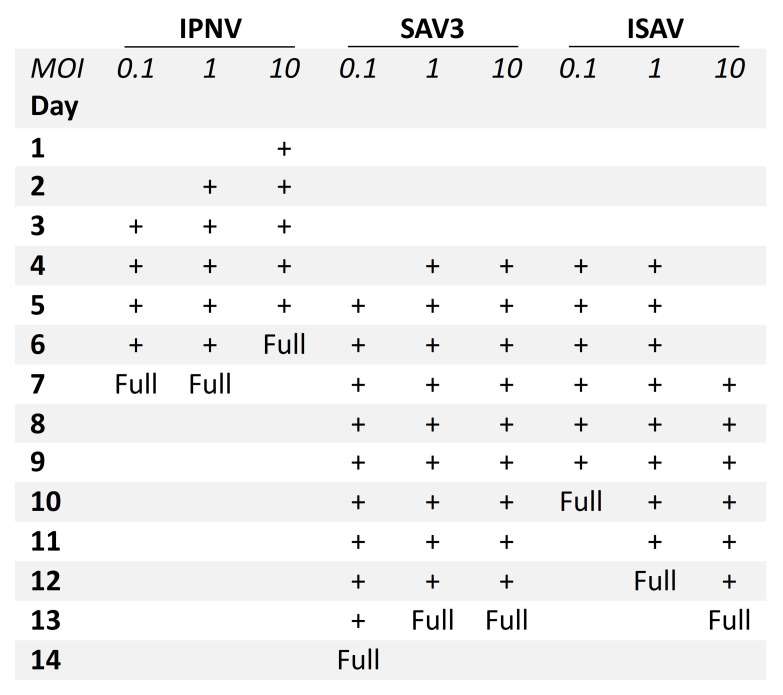
Progression of CPE in RTgutGC cells infected with IPNV, SAV3 and ISAV. Three different multiplicities of infection (MOI) were tested (0.1, 1 and 10), and CPE from start to full CPE (Full) is indicated with +.

**Figure 3 cells-12-01531-f003:**
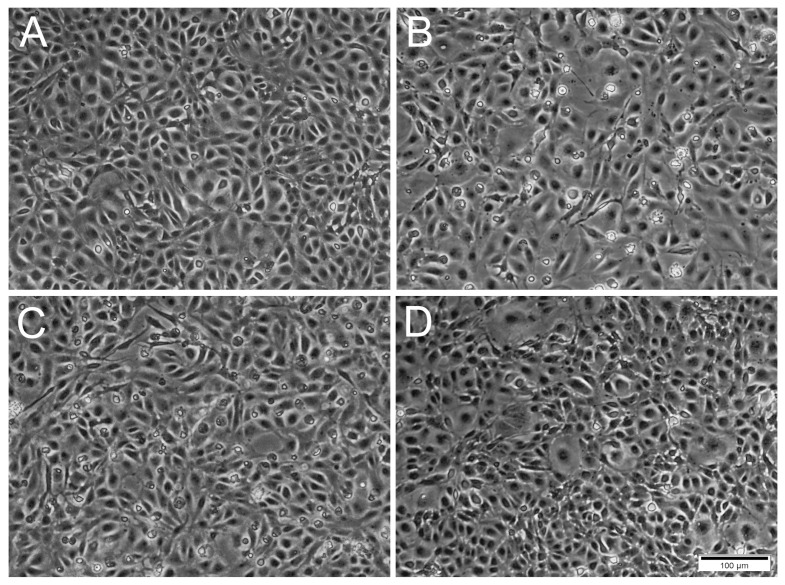
Phase contrast microscopical images of RTgutGC cells inoculated with ISAV at different MOI at 5 days post-infection. (**A**) Uninfected cells. (**B**,**C**) Cells inoculated with ISAV at MOI 0.1 and 1, respectively, with changed cell morphology and presence of rounded dead cells. (**D**) Cells inoculated with MOI 10 resembling control cells. Scale bar 100 µm.

**Figure 4 cells-12-01531-f004:**
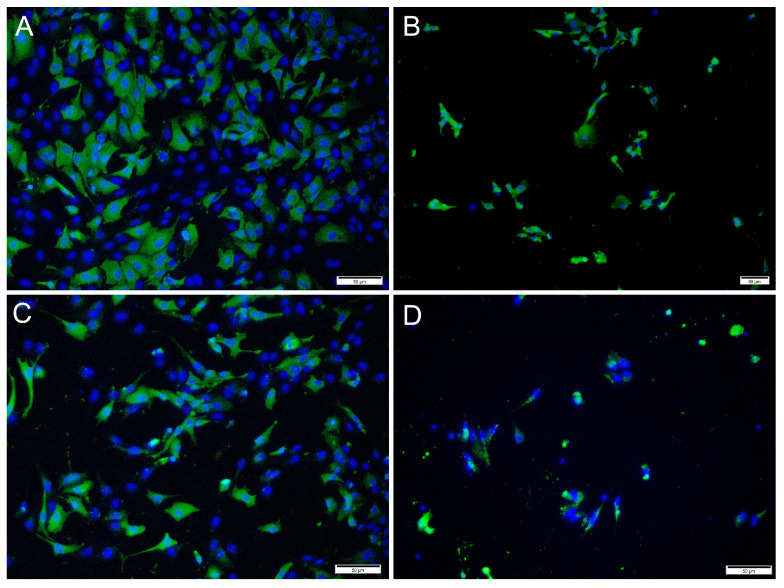
Microscopical images of immunofluorescence staining of RTgutGC cells infected with IPNV. The figure illustrates the detection of IPNV (green) in cells inoculated with MOI 1 (**A**,**B**) and MOI 10 (**C**,**D**) at 2 (left column) and 7 (right column) dpi, respectively. Nuclei are stained blue. Scale bars 50 µm.

**Figure 5 cells-12-01531-f005:**
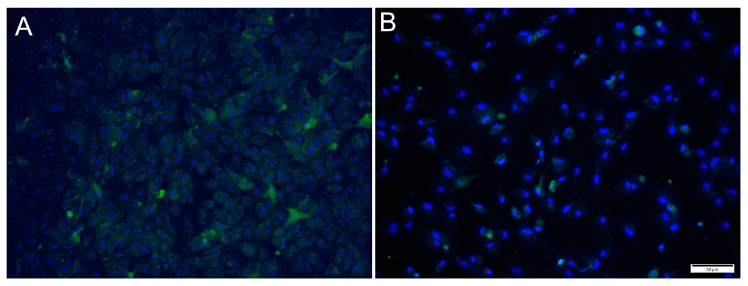
Microscopical images of immunofluorescence staining of RTgutGC cells inoculated with SAV3. The figure illustrates the detection of SAV3 (green) in cells infected with SAV3 at MOI = 10 at 6 dpi (**A**) and MOI = 1 at 14 dpi (**B**). Nuclei are stained blue. Scale bar 50 µm.

**Figure 6 cells-12-01531-f006:**
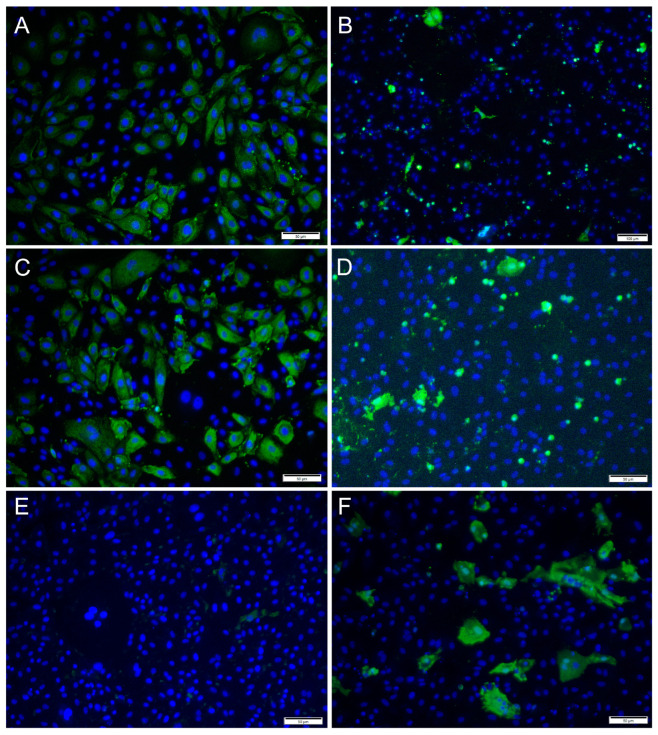
Immunofluorescence staining of RTgutGC cells inoculated with ISAV. The figure illustrates the detection of ISAV (green) in cells infected with ISAV at MOI = 0.1 (**A**,**B**), MOI = 1 (**C**,**D**) and MOI = 10 (**E**,**F**) at 2 (left column) and 7 (right column) dpi, respectively. Nuclei are stained blue. Scale bars (**A**,**C**–**F**) 50 µm, (**B**) 100 µm.

**Figure 7 cells-12-01531-f007:**
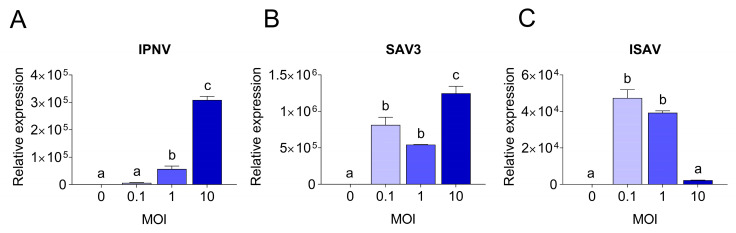
Quantitative PCR results showing viral yields in RTgutGC cells. Infection with IPNV (**A**), SAV3 (**B**) and ISAV (**C**) at MOI 0.1, 1 or 10, and in uninfected cells (MOI = 0). For IPNV, cells were harvested at 3 dpi, while for SAV3 and ISAV, cells were harvested at 5 dpi. Letters indicate significant differences between different MOI.

**Figure 8 cells-12-01531-f008:**
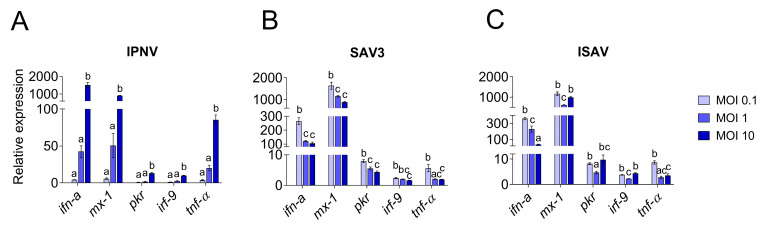
Quantitative PCR expression of antiviral response genes in RTgutGC cells. Infection with IPNV (**A**), SAV3 (**B**) and ISAV (**C**) at MOI 0.1, 1 or 10. Gene expressions are given as relative values to the control group (=1). Letters indicate significant differences between the different MOI where the letter a is insignificant from control.

**Figure 9 cells-12-01531-f009:**
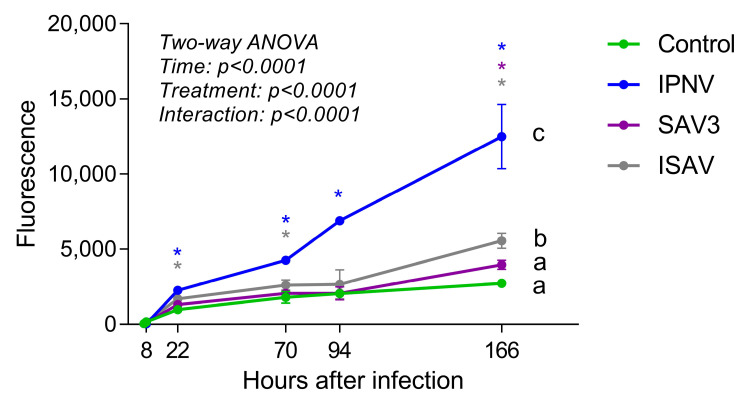
Barrier permeability assay with fluorescent bovine serum albumin (BSA). Cells were incubated with virus (IPNV and SAV3, MOI 1; ISAV, MOI 0.1) for 6 h, fluorescent BSA was then added to the apical well side and samples taken from the basolateral side at different time points. Different letters indicate significant differences between treatments (Tukey’s multiple comparison test). Stars indicate significant differences from control wells for each time point (Dunnett’s multiple comparison test).

## Data Availability

The data presented in this study are available on request from the corresponding author.

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
