# Peer review of "Establishment of an In Vitro Model to Study Viral Infections of the Fish Intestinal Epithelium"

_cells, 2023, doi:10.3390/cells12111531_

Round 1

Reviewer 1 Report

This manuscript is a follow-up study of their previously published paper. The authors used a well characterized RTgutGC cell line as a model system to study the viral infection in vitro. They assessed the infection of infectious pancreatic necrosis virus (IPNV), salmonid alphavirus 3 (SAV3) and infectious salmon anemia virus (ISAV), and reported that the replication kinetics and ability to induce CPE and host responses are different. In addition, the permeability assays showed that the epithelial barrier function was negatively affected by the viral infections before CPE could be detected, and the replication of IPNV and ISAV had a more significant effect on barrier function.

In general, the manuscript is well written and I didn’t find logic flaws by reading it. The methods were described clearly, and the results were scientifically interpreted. There are only a few minor comments.

1)    All micrographs are missing scale bars.

2)    Figure 2 is slightly confusing. I wonder if there is a better way to present it.

3)    The authors compared images from different MOIs in Figure 5. A was taken at MOI=10 at 6 dpi, B was taken at MOI=1 at 14 dpi. I would suggest keeping the same format as Figure 4 and Figure 6. In other words, comparing images at the same MOI but different dpi.

The language is overall very good. Would need a double check on grammars.

Author Response

The authors thank you for your positive response.

1) Thank you for pointing out that scale bars were missing. These have now been added to the figures. Also, we discovered that the order of the pictures was incorrect in figure 4, which has now been corrected as well as correct reference to the figure in the text, lines 301-304.

2) Figure 2 has been replaced with a figure with a different design, hopefully to make these results clearer.

3) We agree to the comment, but due to faint staining of SAV3 as mentioned already in the discussion part (line 486-488), we do not have pictures from all MOIs corresponding to the pictures presented for IPNV and ISAV (Figure 4 and 6), and hope that the editor and reviewers can accept to keep Figure 5 as it stands.

Comments on the Quality of English Language

Some grammatical errors have been corrected throughout the manuscript.

Reviewer 2 Report

The authors used the established cell line RTgutGC to investigate the permissiveness towards IPNV, SAV3, and ISAV. They found all selected viruses replicated in RTgutGC cells. This cell lines provide an alternative tool for viral research. However, comparing with organoid, it might be improper to call RTgutGC as an “model”.

 My comments listed as following:

(1) The authors could try 3D culture of RTgutGC, which might be used as an

“model”.

(2) what is the difference between RTgutGC and other permissive cell lines towards IPNV, SAV3, and ISAV, for instance the titer, one-step amplification curve……

(3) rewriting the section of Abstract. The main contents should be covered.

(1) rewriting the section of Abstract. The main contents should be covered.

(2) Shorten the section of Introduction.

Author Response

Thank you for constructive comments. We have addressed your comments below.

We use the term “in vitro fish intestinal infection model” when referring to combining the transwell system with RTgutGC cells and the selected salmonid viruses. Here, we gained knowledge on the permissiveness of the RTgutGC cells to these viruses as well as identified the suitable cell-to-virus ratio and time points for investigating barrier function. This makes it possible to use the presented set-up or “model system” further to generate more knowledge about the infection pathways and mechanisms used by these viruses to enter the host.

1) To test the RTgutGC cell line as an organoid cluster is an interesting suggestion, and something we should try. However, in this case, to study barrier functions, a system where the intestinal cells are grown on a membrane insert that separates an apical and a basolateral chamber was beneficial. With such system, the luminal surface is highly accessible, and continuous samplings can be made from both apical (“lumen”) and basolateral (“blood”) compartments to assess transport across the intestinal epithelial barrier. In mammals, intestinal organoid models have been developed from self-organizing stem cells to better mimic the 3D structure of the intestine. However, the cystic and spherical structure of these organoids have limitations in accessing the luminal compartment as this will be buried within a Matrigel mass.

Further, in mammals, most organoid intestinal models have been derived from isolated intestinal stem cells expressing Lgr5. In salmonid fish, the stem cell niche has just started to be unraveled, but isolation and culturing of isolated cells from these niches has so far not been successful. Lgr5 was found to be rarely expressed in the rainbow trout intestine in vivo and only in cells of the lamina propria while never in the epithelium, indicating that salmonid fish has a different intestinal stem cell niche compared to mammals with sox9 expression as the main stem cell marker.

Discussion of the transwell set-up used in this study versus organoid models have been included in the manuscript, lines 436-456.

2) This is of course an interesting suggestion, but direct comparisons with other cell lines concerning titers and cell responses lies beyond the scope of this study. To make such comparisons, controlled experiments must be performed with identical conditions. With this study, the aim to test the permissiveness and responses of this particular cell line to the selected viruses, but comparisons with other cell lines is something we could consider doing in the future.

3) Some changes have been made to the abstract to cover main content of the paper, lines 24-27.

Comments on the Quality of English Language

1) Se comment above

2) Thank you for the comment, we agree that the introduction contains a lot of information, but we have decided not to shorten the introduction section as we think this information is beneficial to the reader for the interpretation of the results and the following discussion.

Reviewer 3 Report

This study, which was created in vitro for viral agents living in fish intestines, has remarkable results in terms of animal welfare. I congratulate those who planned and implemented the study.

Author Response

We thank you for your positive and kind response. Some changes have been made to the manuscript according to the comments from the editor and the other reviewers, and we hope you agree to these.